# Effect of Crosslinking Conditions on the Transport of Protons and Methanol in Crosslinked Polyvinyl Alcohol Membranes Containing the Phosphoric Acid Group

**DOI:** 10.3390/polym15214198

**Published:** 2023-10-24

**Authors:** Zhiwei Wang, Hao Zheng, Jinyao Chen, Wei Wang, Furui Sun, Ya Cao

**Affiliations:** 1State Key Laboratory of Polymer Materials Engineering, Polymer Research Institute, Sichuan University, Chengdu 610065, China; kekedope@163.com (Z.W.); zhenghaodwyyx@163.com (H.Z.); caoya@scu.edu.cn (Y.C.); 2Science and Technology on Advanced Functional Composites Laboratory, Aerospace Research Institute of Materials and Processing Technology, Beijing 100076, China; 13552958862@163.com (W.W.); sunfurui1202@163.com (F.S.)

**Keywords:** polyvinyl alcohol, crosslinking, direct methanol fuel cell, proton conductivity, methanol permeability

## Abstract

In this investigation, we systematically explored the intricate relationship between the structural attributes of polyvinyl alcohol (PVA) membranes and their multifaceted properties relevant to fuel cell applications, encompassing diverse crosslinking conditions. Employing the solution casting technique, we fabricated crosslinked PVA membranes by utilizing phosphoric acid (PA) as the crosslinking agent, modulating the crosslinking temperature across a range of values. This comprehensive approach aimed to optimize the selection of crosslinking parameters for the advancement of crosslinked polymer materials tailored for fuel cell contexts. A series of meticulously tailored crosslinked PVA membranes were synthesized, each varying in PBTCA content (5–30 wt.%) to establish a systematic framework for elucidating chemical interactions, morphological transformations, and physicochemical attributes pertinent to fuel cell utilization. The manipulation of crosslinking agent concentration and crosslinking temperature engendered a discernible impact on the crosslinking degree, leading to a concomitant reduction in crystallinity. Time-resolved attenuated total reflection Fourier transform infrared spectroscopy (ATR-FTIR) was harnessed to evaluate the dynamics of liquid water adsorption and ionomer swelling kinetics within the array of fabricated PVA films. Notably, the diffusion of water within the PVA membranes adhered faithfully to Fick’s law, with discernible sensitivity to the crosslinking conditions being implemented. Within the evaluated membranes, proton conductivities exhibited a span of between 10^−3^ and 10^−2^ S/cm, while methanol permeabilities ranged from 10^−8^ to 10^−7^ cm^2^/s. A remarkable revelation surfaced during the course of this study, as it became evident that the structural attributes and properties of the PVA films, under the influence of distinct crosslinking conditions, underwent coherent modifications. These changes were intrinsically linked to alterations in crosslinking degree and crystallinity, reinforcing the interdependence of these parameters in shaping the characteristics of PVA films intended for diverse fuel cell applications.

## 1. Introduction

In recent decades, the burgeoning global population and rapid industrial advancement have accentuated the profound challenges of environmental pollution and the depletion of finite natural resources. Fossil fuels, serving as a prominent energy source, have been associated with the emission of hazardous pollutants into the atmosphere [1,2]. Furthermore, the sustainability of fossil fuel reserves has raised mounting apprehensions, as their eventual depletion is an inevitable prospect [3,4,5]. Against this backdrop, proton exchange membrane fuel cells (PEMFCs) have emerged as a compelling contender to supplant conventional fossil fuels across a spectrum of applications, spanning from portable devices to electric vehicles. The attractiveness of PEMFCs stems from their compelling attributes such as environmental benignity, elevated energy density, minimized emissions, and expedited low-temperature activation [6]. At the heart of these advancements lies the polymer electrolyte membrane (PEM), which facilitates proficient proton conduction while demarcating the anode and cathode compartments [7].

Proton exchange membranes harnessed within PEMFCs are primarily founded upon aliphatic perfluorinated polymers integrated with sulfonic acid moieties [8]. Pioneered by DuPont in the late 1960s, the Nafion^®^ membrane, an exemplar of a perfluorinated sulfonated ionomer, has solidified its stature as a quintessential PEM material. This eminence is attributed to its relatively robust proton conductivity (∼0.1 S/cm) under fully hydrated conditions, coupled with commendable mechanical robustness and thermal stability. Nafion^®^ membrane excels particularly in fuel cell operations at moderate temperatures below 90 °C and elevated relative humidity, with hydrogen as the fuel source [9,10]. Notwithstanding Nafion’s ubiquity in fuel cell applications, its widespread adoption is curtailed by its prohibitive fabrication costs and its diminished conductivity in anhydrous conditions at elevated temperatures due to moisture evaporation [11]. Notably, a critical drawback of perfluorinated ionomers, particularly pertinent to direct methanol fuel cell (DMFC) deployment, is their pronounced methanol permeability (∼10^−6^ cm^2^/s) [12]. This phenomenon, recognized as methanol crossover, not only curtails methanol utilization efficiency but also substantially exacerbates the cathodic oxygen electrode’s polarization. Consequently, this undermines the performance of direct methanol fuel cells [13,14].

Polyvinyl alcohol (PVA) assumes a pivotal role as a significant chemical precursor, finding wide-ranging applications in the production of polyvinyl acetals, gasoline pipelines, vinylon synthetic fibers, fabric primers, and more [15]. Its molecular structure is characterized by an abundance of hydroxyl groups, fostering facile hydrogen bond formation between molecules. Notably, PVA stands distinguished as an exemplary membrane material, renowned for its commendable film-forming attributes, non-toxicity, heightened hydrophilicity, biocompatibility, and remarkable chemical resilience in the presence of organic solvents [16,17]. Additionally, the exceptional dielectric permittivity and potent charge storage capacity of PVA prompted its selection as a supercapacitor material [18].

Remarkably, PVA membranes exhibit a propensity to effectively curb methanol permeability, outperforming their Nafion counterparts. Evidencing the discerning water-to-alcohol selectivity inherent in commercial PVA membranes, Cussler and collaborators [19] delineated the superior methanol barrier characteristics of PVA membranes employed in pervaporation processes, surpassing Nafion^®^ membranes. Of particular note, PVA membranes have found application in ethanol dehydration, circumventing the ethanol–water azeotrope, owing to their preferential passage of water molecules over ethanol or methanol [20,21,22]. However, PVA inherently lacks any ionic conducting moieties, such as carboxylic and sulfonic acid groups. To address this limitation, Ebenezer et al. [23]. introduced a promising avenue wherein PVA membranes were crosslinked via a transesterification process utilizing sulfosuccinic acid (SSA), a crosslinking agent with sulfonic acid functionality, thereby imparting prospective proton conductivity to the PVA membrane.

While sulfonic acid, a hallmark functional group in Nafion membranes, exhibits robust proton dissociation and conduction capabilities under fully hydrated conditions, its capacity to bind protons diminishes considerably at low humidity and temperature levels. Additionally, the proton conduction of sulfonic acid groups is inherently contingent upon water participation, leading to decreased proton conductivity at high temperatures and low humidity. In contrast, phosphate emerges as an alternative proton transfer site within proton exchange membranes [24]. The amphoterism of phosphoric acid, endowing it with both proton donating and accepting abilities, enables the formation of a dynamic hydrogen bond network within the membrane. This network facilitates proton transmission through the dynamic breaking and reformation of hydrogen bonds between molecules. Importantly, phosphate groups exhibit potent proton self-dissociation ability, even in the absence of water, and feature a low free energy barrier for proton transfer between phosphate groups. Moreover, phosphoric acid boasts heightened thermal stability and oxidation resistance compared to sulfonic acid. Hence, phosphonic acid has garnered substantial attention for crafting proton exchange membranes suitable for high-temperature, low-humidity PEMFCs [25,26].

Despite these advances, the realm of phosphorylated membranes for fuel cell applications remains underexplored. Thus, in the present study, we harnessed PBTCA, a phosphoric acid-bearing compound, as both a crosslinking agent and a proton transfer medium. This dual role was designed to bestow reasonable proton conductivity to PVA membranes in fuel cell contexts. The incorporation of negatively charged ion groups into the PVA matrix was achieved through chemical modification via crosslinking with PBTCA. Notably, the immobilization of phosphate groups transpired through the esterification reaction between -COOH groups on PBTCA and -OH groups on PVA [27]. In conclusion, this endeavor aspires to assess the viability of crosslinked PVA membranes harboring phosphate groups as promising PEM candidates for fuel cell applications. Moreover, through methodically altering the crosslinking parameters, the investigation delves into the inherent interconnection between crystalline crosslinked membrane architectures and their attendant performance metrics, thus illuminating a path toward enhanced fuel cell performance.

## 2. Experimental

### 2.1. Materials

Polyvinyl alcohol (PVA, 99% hydrolyzed, with a polymerization degree of 2400) was supplied by the ichuan vinylon factory of Sinopec (Chongqing, China). Briefly, 2-phosphonobutane-1,2,4-tricarboxylic acid (PBTCA) was purchased from Shanghai Macklin Biochemical Co., Ltd., Shanghai, China. The deionized water was self-made.

### 2.2. Membrane Preparation

PVA/PBTCA composite membranes were fabricated using the solution casting technique. Specifically, 3 g of PVA was dissolved in 27 mL of deionized water and stirred at 90 °C for several hours until complete dissolution was achieved, yielding a clear and transparent PVA solution with a concentration of 10 wt.%. The pH of the solution was adjusted to 1 through the addition of H_2_SO_4_. Subsequently, various quantities by the weight of PBTCA (ranging from 5 to 30 wt.%) were introduced into the solution and stirred at 50 °C for a duration of 4 h. The resulting solution was then poured into a PTFE mold measuring 13 × 13 cm, where it was subjected to drying until complete solvent evaporation was achieved. The film was peeled from the mold and heat-treated in an oven. The resulting samples were designated as PVA/PBTCA-X-Y, where X represents the PBTCA content, and Y represents the crosslinking temperature. The reaction process between PBTCA and PVA is illustrated in Figure 1.

### 2.3. Membrane Characterization

The chemical structures of the original and cross-linked PVA films were elucidated through Fourier transform infrared spectroscopy (FTIR). Each sample underwent 16 scans, employing a spectral resolution of 4 cm^−1^. Simultaneously, the physical morphology of the fabricated films was assessed via scanning electron microscopy (SEM). The degradation process and the thermal stability of membranes were evaluated using Thermogravimetric analyzer (JEM-F200, JEOL, Tokyo, Japan). The membranes sample were subjected to heat from 25 to 600 °C with a heating rate of 10 °C/min. The crystallization structure of membranes was observed using a X-ray diffractometer (tall Ultima IV, Rigaku Co., Tokyo, Japan) with a scanned area in the range of 5° ≤ 2θ ≤ 50°.

#### 2.3.1. Water Uptake, Hydrolytic Stability, and Crosslinking Density

The membrane specimens were immersed in deionized water for a duration of 24 h. Post-immersion, the membrane samples were retrieved, and surface moisture was meticulously blotted, whereupon their weights were recorded as W_d_. Subsequently, these membrane samples were subjected to desiccation in a vacuum oven at 80 °C until a consistent mass was reached, and their masses were re-determined and denoted as W_w_. The water uptake of membranes can be calculated via Equation (1):(1)$$Water\;uptake\left(\%\right)=\frac{W_{w}-W_{d}}{W_{d}}*100$$

Equation (2) was used to calculate the hydrolytic stability of crosslinked PVA-based films.
(2)$$Weight\;loss=\left(1-\frac{W_{a}}{W_{b}}\right)\times 100$$
where W_a_ is the weight after drying following 80 °C drip washing, and W_b_ is weight before drying following 80 °C drip washing.

The crosslinking density of a crosslinked polymer is expressed by the number of moles of polymer fragments between adjacent crosslinking points, and calculated using the following formula: (3)$$P=\frac{1}{vM_{C}}$$
where v is the specific volume of PVA (=0.788 mol cm^−3^) and M_c_ is the molecular weight (kg mol^−1^) between the crosslink sites calculated using Equation (2) deduced from the Flory–Rehner relation [28]: (4)$$Mc=\frac{V_{S}\left[\phi^{\frac{1}{3}}-\frac{1}{2}\phi \right]}{\left[\ln\left(1-\phi \right)+\phi +\chi \phi 2\right]}$$
where V_S_ is the molar volume of the water (cm^3^ mol^−1^), 𝜌_ρ_ is the density of the crosslinked PVA-based films (=1.33 g cm^−3^), and χ is the interaction parameter of PVA–water (=0.494); 𝜙 is the volume fraction of polymer in a water-swelled sample and can be calculated via Equation (5): (5)$$\phi =\frac{\left(w_{d}\right){\rho_{p}}^{-1}}{\left[\left(w_{d}\right){\rho_{p}}^{-1}+\left(w_{s}\right){\rho_{s}}^{-1}\right]}$$
where 𝑊_𝑑_ is the weight of the dry crosslinked PVA-based film (g); 𝑊𝑠 is the weight of absorbed water in swollen PVA-based film (g); and 𝜌_𝑠_ is the density (g cm^−3^) of the solvent (water).

#### 2.3.2. Time-Resolved FTIR-ATR Spectroscopy

Time-dependent infrared spectroscopy, facilitated by an attenuated total reflection fourier transform spectrometer (Frontier FT-IR Spectrometer, PerkinElmer Co., Ltd., Waltham, MA, USA), was employed to investigate the diffusion of liquid water and the dynamics of polymer swelling within PVA-based films. Prior to the initiation of the transmission experiment, background spectra of the ATR crystal were recorded, and each subsequent spectrum collected was differentially processed by subtracting the background spectrum for calibration. The PVA film was directly cast onto a ZnSe optical window, followed by drying in a high-purity nitrogen environment. Subsequently, a layer of hydrophilic polytetrafluoroethylene (PTFE) filter paper was overlaid onto the sample. Two scenarios were investigated for water molecule diffusion, one involving dry PVA-based films and the other utilizing deionized water. For the latter, deionized water was introduced into the filter paper, and spectra were collected. The ATR cell was sealed during the experiment to prevent water evaporation. Time-resolved ATR-FTIR data were collected in one-minute intervals until spectral stability was achieved, with each spectrum being averaged over 16 scans at a resolution of 4 cm^−1^. Post-experiment, the thickness of the hydrated film was immediately measured, with each thickness value representing the average of measurements taken at five distinct positions on the film.

Infrared light penetration into the sample exhibited a thickness of approximately 0.5 µm, which is significantly less than the film’s thickness. Moreover, the rate of water spreading on the PTFE filter exceeded its diffusion velocity in the PVA film thickness direction. As such, we conclude that water diffusion in the film thickness direction exhibited uniformity.

In this diffusion experiment, the water diffusion kinetics within PVA-based films were aptly described by Fick’s second law, employing the one-dimensional continuous equation:(6)$$\frac{\partial C}{\partial t}=D\frac{\partial^{2}C}{\partial z^{2}}$$
where C denotes the water concentration, z represents the distance traversed through the film (with z = 0 at the polymer–ATR interface and z = L at the polymer–water interface), t signifies time, and D stands for the effective average diffusion coefficient of water within the polymer.

The initial and boundary conditions for the one-dimensional water transport through PVA are as follows:(7)$${C=C}_{0}\;at\;0<z<L\;and\;t=0$$
(8)$$C=C_{L}\;at\;z=L\;and\;t≧0$$
(9)$$\frac{dC}{dz}=0\;at\;z=0\;and\;t\geq 0$$

Here, C_0_ signifies the initial water concentration within the membrane (assumed to be zero in this study), while C_L_ denotes the water concentration at the membrane’s surface. To facilitate the description of the experimental setup and data analysis, a specific choice of coordinates was employed. The origin (z = 0) was designated as the interface between the polymer and the ATR crystal, with z = L representing the interface between the polymer and water. The analytical solution to Equation (6), considering these initial and boundary conditions, is as follows:(10)$$\frac{C\left(t\right)-C_{0}}{{C_{eq}-C}_{0}}=1-\frac{4}{\pi }\sum_{n=0}^{\infty }\frac{{\left(-1\right)}^{n}}{2n+1}\exp\left(-Df^{2}t\right)\cos\left(fz\right)$$
$$f=\frac{(2n+1)\pi }{2L}$$

The parameters C(t) and C_eq_ correspond to the water concentration at any specific time, t, and the steady-state equilibrium water concentration within the film, respectively. Employing the differential form of the Beer–Lambert law, the relationship between infrared absorbance and concentration can be expressed as follows:(11)$$A=\int_{0}^{L}\varepsilon^{*}C\left(t\right)\exp\left(\frac{-2z}{d_{p}}\right)dz$$

In this equation, A stands for the ATR absorbance, ε* represents the molar extinction coefficient, and d_p_ indicates the depth of penetration of the evanescent wave within the polymer. This penetration depth is influenced by the refractive indices of both the polymer and crystal material.
(12)$$d_{p}=\frac{\lambda }{2\pi n_{1}\sqrt{\sin^{2}\theta -{\left(n_{2}/n_{1}\right)}^{2}}}$$

Here, n_1_ and n_2_ denote the refractive indices of the ATR crystal and PVA, respectively (n_1_ = nZnSe = 2.4; n_2_ = nPVA = 1.5), θ signifies the angle of incidence of the light, and λ denotes the wavelength of the absorbing light. In the context of this study, d_p_ ≈ 1 µm. To fit the experimental data, Equation (10) can be substituted into Equation (11) and integrated, leading to Equation (13):(13)$$\frac{A\left(t\right)-A_{0}}{{A_{eq}-A}_{0}}=1-\left\{\frac{8}{\pi d_{p}\left[1-exp\left(\frac{-2L}{d_{p}}\right)\right]}\right\}\times \sum_{n=0}^{\infty }\frac{1}{2n+1}\left\{\frac{\exp\left(-Df^{2}t\right)[fexp\left(\frac{-2L}{d_{p}}\right)+{\left(-1\right)}^{n}(\frac{2}{d_{p}})}{({\frac{2}{d_{p}})}^{2}+f^{2}}\right\}$$
where A(t), A_0_, and A_eq_ represent the ATR absorbance values at any given time, t, at t = 0, and at equilibrium, respectively. Given the substantial film thickness relative to the penetration depth of the evanescent wave (L/d_p_ > 10), Equation (10) effectively approximates Equation (13). Consequently, as the concentration at the interface between the polymer and ATR crystal (z = 0) remains nearly constant, the following solution is derived:(14)$$\frac{A\left(t\right)-A_{0}}{{A_{eq}-A}_{0}}=\frac{C\left(t\right)-C_{0}}{{C_{eq}-C}_{0}}=1-\frac{4}{\pi }\sum_{n=0}^{\infty }\frac{{\left(-1\right)}^{n}}{2n+1}exp(-Df^{2}t)$$

The penetration of water into the polymer film induces osmotic stresses, resulting in the polymer undergoing swelling. The time-dependent behavior of this swelling process, arising from water diffusion, can be suitably described using a triexponential viscoelastic model [29,30], expressed via the following equation:(15)$$\varepsilon \approx \frac{A_{0}}{A\left(t\right)}-1=\frac{\sigma_{0}}{\eta }t+\frac{\sigma_{0}}{E}\left[1-\exp\left(-\beta t\right)\right]$$
where σ_0_, ε, η, E, and β represent stress, strain, dynamic viscosity, Young’s modulus, and the relaxation time constant, respectively. 

The coupled diffusion–relaxation model has been intricately formulated by integrating Equations (14) and 15, thereby facilitating the dissection of water diffusion in the polymer into two distinct contributions, one driven by the concentration gradient of water, and the other arising from additional water absorption due to polymer swelling and relaxation [31].
(16)$$\frac{A\left(t\right)-A_{0}}{{A_{eq}-A}_{0}}=F_{A}\left[1-\frac{4}{\pi }\sum_{n=0}^{\infty }\frac{{\left(-1\right)}^{n}}{2n+1}\exp\left(-Df^{2}t\right)\right]+F_{B}\left\{w_{2}\frac{t}{t_{f}}+w_{1}\left[1-\exp\left(-\beta t\right)\right]\right\}$$
$$w_{1}=\frac{\eta /E}{t_{f}+\eta /E}$$
$$w_{1}=\frac{t_{f}}{t_{f}+\eta /E}$$

In this context, w_1_ and w_2_ are normalized in accordance with Equation (15), while F_A_ and F_B_ denote the weight fractions of Fickian diffusion and polymer swelling contributions to the overall water uptake kinetics. The water diffusion coefficient, D, as the sole adjustable parameter, can be accurately determined by fitting the time-resolved absorbance data to Equation (16).

In summary, this comprehensive model provides an insightful description of water diffusion and polymer swelling dynamics within PVA films, allowing for the determination of the water diffusion coefficient and the precise quantification of individual contributions from these diverse processes to the overall water uptake kinetics.

#### 2.3.3. Free Volume Testing

Positron annihilation spectroscopy was employed to probe the free volume characteristics within the polymer [32]. When positrons are introduced into the polymer sample, they undergo annihilation with electrons. This annihilation process involves two possible pathways: direct annihilation with electrons and the formation of bound states known as positronium (Ps). Notably, there are two types of Ps: ortho-positronium (o-Ps) and para-positronium (p-Ps), constituting 75% and 25% of the total proportion, respectively. Within the free volume regions of the material, o-Ps undergoes multiple collisions with the pore walls in the free volume cavities, subsequently capturing an electron with an opposite spin orientation on the pore wall before “picking up” annihilation occurs. Notably, this process has no influence on p-Ps, but it does result in a reduced annihilation lifetime for o-Ps, diminishing from 142 nanoseconds to mere nanoseconds. The annihilation lifetime of o-Ps serves as a significant indicator of the free volume size in the material. By precisely measuring the lifetime and relative strength of o-Ps, valuable information concerning the free volume size and free volume fraction of the material can be deduced [33,34,35]. 

The radius of the pores between the molecular chains of PVA-based films can be calculated via Equation (17):(17)$$\tau_{3}={\frac{1}{2}\left[1-\frac{r}{r+\Delta r}+\frac{1}{2\pi }\sin\left(\frac{2\pi r}{r+\Delta r}\right)\right]}^{-1}$$
where r is the average radius of micropores between polymer molecular chains, and Δr = 0.166 nm is an empirical value estimated from a material with a known free pore size.

#### 2.3.4. Proton Conductivity Testing 

Methanol permeability was quantified utilizing the double diffusion cell method. The protonic conductivity of the membrane was ascertained using a four-electrode AC impedance method, implemented on an electrochemical workstation. Sample specimens were tailored to dimensions of 4 cm in length and 1 cm in width, within a scanning frequency spectrum spanning from 0.1 Hz to 1 MHz. The proton conductivity is calculated via the following Equation (18):(18)$$\sigma =\frac{L}{AR}$$
where σ is the proton conductivity (S·cm^−1^); L is the thickness of the film (cm); A denotes the effective contact area of the film (cm^2^); R is the resistance of the film (Ω).

#### 2.3.5. Methanol Permeability Test

The membrane samples were precision-cut into circular discs with a radius of 1 cm and were securely positioned between two diffusion cells. Diffusion cell A was charged with 150 mL of 2 M methanol solution, while diffusion cell B received 150 mL of deionized water. Each diffusion cell was treated independently with continuous stirring. Throughout the experimental procedure, discrete aliquots of methanol solution were extracted from the cell A in 30 min intervals, and their concentrations were quantified via gas chromatography. Four such measurements were conducted to establish the temporal evolution of methanol concentration. The calculation of methanol permeability (P, cm^2^·s^−1^) can be expressed as follows:(19)$$P=\frac{S\;V_{B}\;L}{A\;C_{A}}$$
where S represents the linear fitting slope of the methanol concentration versus time curve; V_B_ corresponds to the volume of solution contained in cell B, which is 30 mL; L denotes the thickness of the film (cm); A is the effective area of the membrane (cm^2^); C_A_ is the initial concentration of methanol in the cell A (2 mol·L^−1^). σ is the proton conductivity (S·cm^−1^); L is the thickness of the film (cm); A represents the effective area of the membrane (cm^2^); R is the resistance of the film (Ω). 

#### 2.3.6. IEC

The ion exchange capacity (IEC), also known as the IEC value, is defined as the milligram equivalent of exchangeable cations (H^+^) per gram of the dry-state ion exchange membrane. In this study, the IEC value of the membranes was determined through the classical titration technique. Initially, the thoroughly desiccated sample is immersed in a 0.1 M HCL solution at 70 °C, facilitating the complete conversion of the film into its H+ form. Subsequently, the meticulously cleansed sample is submerged in a saturated NaCl solution for a duration of 24 h, ensuring the complete release of H+ ions into the sodium chloride solution. This sodium chloride solution is subsequently titrated to neutrality with a 0.1 M NaOH solution, employing phenolphthalein as the indicator. The IEC value (mmol/g) was calculated via the following equation: (20)$$IEC=\frac{M_{O,NaOH}-M_{E,NaOH}}{W_{dry}}$$
where M_O,NaOH_ is the milli-equivalent (meq.) of NaOH in the flask at the beginning of the titration, M_E,NaOH_ is the meq. of NaOH after equilibrium, and W_dry_ is the weight of the dry membrane (g).

#### 2.3.7. Single Cell Performance

In terms of the preparation of MEA, MEA was prepared via the CCM (catalyst-coated membrane) method. Pt-Ru/C was selected as the anode catalyst and JM60% Pt/C as the cathode catalyst. The anode and cathode catalysts were dispersed in 5 wt.% isopropyl alcohol solution and ultrasonic treatment for 30 min to obtain the anode and cathode catalysts. The mass ratio of catalyst to isopropyl alcohol was 5:1, and the binder bone content of catalyst layer was 28%. Then, the catalyst was sprayed on the 280 μm carbon paper diffusion layer with a spray gun. The catalyst load at the two poles was 3 mg cm^−2^, and the active area of the MEA was 6 cm^2^. Then, the film to be measured was sandwiched in the middle of the catalyst layer, and hot-pressed for 4 min at 100 °C and 3 MPa; finally, the MEA was obtained successfully. 

Utilized the AL-CS-150 fuel cell model for the DMFC evaluation. A meticulously prepared MEA was integrated into the fuel cell testing apparatus. Testing parameters were set at 60 °C with a relative humidity of 40%. The anode was supplied with a 2 M methanol solution at a flow rate of 3 mL min^−1^. Additionally, the cathode was humidified and provided with an oxygen flow rate of 200 mL min^−1^.

## 3. Results and Discussion

### 3.1. Structural Characterization of PVA-Based Crosslinked Films

The surface and cross-section morphologies of the crosslinked PVA/PBTVA membranes are depicted in Figure 2. The micrographs distinctly illustrate that the membranes present a smooth and relatively homogeneous surface, devoid of apparent phase separation and structural defects. The composite membranes exhibit an approximate thickness of 58 μm.

The obtained FT-IR spectra are displayed in Figure 3. The absorption band attributed to the ester group (-COO-) emerges at 1707 cm^−1^. Notably, the intensity of the -C=O- stretching mode within the ester bond increases proportionally with the quantity of PBTCA, indicative of PBTCA’s role in crosslinking PVA polymers via the esterification of hydroxyl groups along the PVA side chain. The absorption band at 1081 cm^−1^ corresponds to the C–OH stretch mode of the ester group. The peaks associated with the phosphoric acid group at 1194 cm^−1^, reflecting the stretching vibrations of P=O, exhibiting enhancement in tandem with increasing PBTCA content. The comprehensive FT-IR results affirm the effective crosslinking and phosphorylation of the PVA matrix through the incorporation of PBTCA, coinciding with an augmented degree of crosslinking. The crosslinking degrees of the membranes under distinct crosslinking conditions are presented in Table 1.

The influence of PBTCA on structural alterations was further elucidated through wide-angle X-ray diffraction. As showcased in Figure 4, the X-ray diffraction patterns of PVA/PBTCA-5-125, PVA/PBTCA-15-125, and PVA/PBTCA-30-125 are displayed. Notably, all samples were observed to exist the robust crystallization peak observed at 2θ = 19.8°, which is a characteristic diffraction feature of PVA. With the augmentation of PBTCA content, a discernible trend is apparent in the diffraction peak of PVA/PBTCA membranes crosslinked at 125 °C—specifically, the peak shape gradually broadens, while the intensity progressively diminishes. The crystallinity of the membranes calculated using Jade v6.0 software undergoes a gradual decline corresponding to the increase in PBTCA content, as indicated in Table 1. This phenomenon arises from the disruption of the ordered structure of PVA molecules via the introduction of PBTCA through esterification reactions involving hydroxyl groups within the main chain. Significantly, this phenomenon reinforces the successful accomplishment of the crosslinking reaction.

The variation in ion exchange capacity (IEC) in the PVA/PBTCA membranes is depicted as a function of PBTCA content (wt.%) in Figure 5. Within the scope of this investigation, the fabricated PVA/PBTCA membranes demonstrated IEC values spanning the interval of 0.65–2.24 mmol/g. The figure reveals a consistent ascending trajectory in the IEC values of the PVA/PBTCA membranes, corresponding to the incremental augmentation of PBTCA content. This phenomenon can be attributed to the inherent presence of phosphoric acid groups within PBTCA, which confer a negative charge. In practical terms, the ion exchange capacity of a membrane offers insights into the abundance of ion-exchangeable groups within it. Consequently, the strategic incorporation of phosphoric acid groups through the utilization of PBTCA emerges as a favorable mechanism for enhancing the IEC values of the membranes.

### 3.2. Stability

A thermal stability analysis of the crosslinked hybrid membranes was conducted through thermogravimetric analysis (TGA). As depicted in Figure 6, the TGA profiles of the membranes manifest three distinctive degradation stages, corresponding to thermal solvation, thermal dephosphorization, and the eventual thermal oxidation of the polymer matrix. The initial phase of thermal weight loss, observed between 80 °C and 120 °C, can be attributed to the evaporation of absorbed water or water by-products resulting from further esterification within the PVA/PBTCA membranes. The subsequent weight loss, at approximately 10%, occurring between 220 °C and 330 °C, is attributed to the decomposition of phosphonic acid groups and the cleavage of ester bonds. In the absence of -SO3H functional groups, pure PVA demonstrates negligible weight loss across this temperature range. Lastly, the third thermal weight loss phase emerges above 400 °C, signifying the degradation of the PVA molecular backbone. As the concentration of PBTCA increases, a reduction in the weight loss of the crosslinked PVA becomes evident, highlighting the reinforcing effect of heightened crosslinking density on thermal stability.

The assessment of PVA’s hydrolytic stability involved subjecting the membrane to droplet washing with deionized water at 80 °C, with the subsequent measurement of weight loss. The outcomes of the drip washing procedure are presented in Figure 7. Weight loss reaches a plateau after 72 h, indicating the completion of the leaching of soluble compounds within the crosslinked membrane. Remarkably, the crosslinked membrane containing 30% PBTCA exhibited the highest water stability, evidenced by the lowest weight loss (5%). Evidently, the crosslinking treatment significantly enhanced the hydrolytic stability of the membrane.

### 3.3. Water Absorption

Figure 8 illustrates the relationship between water absorption and the crosslinking condition controlled via the content of the crosslinking agent and crosslinking temperature. The water absorption rate of the crosslinked membrane exhibited a progressive augmentation until the PBTCA content reached 15 wt.%, signifying that the incorporation of that PBTCA crosslinker into the PVA matrix induced heightened membrane compactness. This structural modification curtailed the mobility of polymer molecules, consequently constraining the ingress and permeation of water molecules within the membrane. Beyond this threshold, the water absorption rate initiated a gradual reduction. This phenomenon stems from the crosslinking process leading to a reduction in polymer crystallinity. As a consequence, the amorphous regions, characterized by a comparatively lax molecular arrangement and heightened mobility, dominate the majority of the film’s spatial extent. This pivotal alteration in architecture contributed to an elevated water absorption propensity. In summation, the confluence of cross-linking density and crystallinity orchestrates a collective influence on the water absorption kinetics of the film, operating through intricate mechanisms of competitive interplay. Figure 9 shows a schematic structure model of the crosslinked PVA/PBTCA membrane involved in the crosslinking temperature and the amount of crosslinking agent.

The water uptake of crosslinked membranes can also be explained using the free volume theory, highlighting the close relationship between the polymer’s free volume and water uptake. The concept of free volume refers to the voids or spaces within polymer molecules, which play a role in facilitating the permeation and diffusion of water molecules. In this experiment, positron annihilation lifetime spectroscopy (PALS) was employed to quantify the free volume of crosslinked PVA-based membranes.

Across the spectrum of membrane samples examined, the observation of three distinctive exponential decay components was discerned. Remarkably, the longest decay component (τ_3_) was linked to the annihilation of ortho-positronium (o-Ps) and was established as a surrogate indicator of the free volume dimensions intrinsic to the selective polymer layer. The test results are shown in Table 2. Surprisingly, upon crosslinking, a coupled variation in free volume and water uptake kinetics was observed. The free volume was similarly influenced by the interplay between crosslinking density and crystallinity. As expounded earlier, heightening the crosslinking density led to a corresponding attenuation in crystallinity, thereby disrupting the alignment of molecular chains and fostering augmented chain mobility. This, in turn, translated to an enlargement of the free volume within the polymer matrix. However, the concurrent augmentation of polymer crosslinking junctions culminated in the establishment of a denser network architecture, consequentially attenuating the free volume.

The interplay between water absorption and free volume stems from the essential role that free volume assumes in creating the requisite spatial capacity for water molecules to permeate and diffuse within the polymer’s architecture. The accessible free volume within the polymer matrix exerts a profound influence on the ease with which water molecules infiltrate and establish a presence within the material’s confines. Broadly speaking, a positive correlation manifests between water absorption and free volume. A larger free volume within the polymer matrix frequently translates to an escalated rate of water absorption. This phenomenon finds its roots in the fact that polymers endowed with an expansive free volume can readily accommodate a heightened quantity of water molecules, facilitated by the augmented availability of open spaces within the material. Consequently, water molecules can effectively permeate the polymer matrix, occupying the available voids and culminating in a more pronounced degree of swelling and water absorption. Conversely, polymers characterized by a more limited free volume tend to exhibit diminished rates of water absorption. The constrained availability of void spaces curtails the diffusion of water molecules and their amalgamation into the polymer matrix. In these scenarios, the polymer’s architecture becomes more tightly knit, presenting fewer pathways for water molecules to traverse and integrate.

To further investigate and validate the penetration and diffusion of water molecules within the crosslinked membranes, a quantitative assessment of liquid water adsorption and swelling kinetics in PVA-based crosslinked films was conducted using time-resolved FTIR-ATR spectroscopy. As illustrated in Figure 10, with the infusion of water into the polymer matrix, a distinct spectral band emerged at 1640 cm^−1^, in concurrence with the bending vibrational mode of hydroxyl groups within the water molecules. The inset within the figure provides an amplified view of the time-resolved O-H stretching region (3500–3000 cm^−1^) and the -CH_2_ stretching region (1422 cm^−1^), denoting the diffusion of water molecules within the polymer and the consequent swelling of the polymer chain instigated by the ingress of water.

Figure 11a displays the regression analysis applied to the polymer-normalized -CH_2_- stretching absorbance data (depicted as red squares), harmonized with the outcomes of the three-element viscoelastic relaxation model (illustrated as a black line, Equation (14)). The dashed black line conveys the linear fit at later stages, signifying the viscous contribution as the exponential term approaches infinity. The exclusive relaxation time constant β, calculated at 9.2 × 10^−5^ s^−1^, emerged as the adjustable parameter. Subsequently, the regressed viscoelastic model parameters were input into the polymer diffusion–relaxation model (Equation (16)), with the diffusion coefficient, D, as the sole adjustable parameter. Figure 11b provides a comprehensive exposition of the outcomes derived from regressing the polymer-normalized O-H stretching absorbance data into the diffusion–relaxation model (Equation (16)), yielding a computed value of D at 1.28 × 10^−6^. Both figures illustrate the accurate capture of polymer swelling kinetics and liquid water molecule diffusion kinetics via the selected three-element viscoelastic model and the diffusion-relaxation model, respectively.

Extending the same analytical methodology to membranes crosslinked at distinct temperatures, the summative findings are encapsulated within Table 3. The congruence between the evolving kinetics of water absorption and swelling in membranes with varying crosslinking densities, as well as the previously investigated water uptake rates and free volume, is distinctly apparent. This conspicuous alignment underscores a nuanced and profound interplay among these tripartite components. Additionally, this alignment serves to substantiate the veracity of the preceding discourse elucidating water uptake rates.

### 3.4. Methanol Permeabilities

The proton exchange membrane (PEM) utilized in direct methanol fuel cells (DMFCs) possesses a crucial and indispensable performance characteristic that profoundly influences the membrane’s operational efficiency and lifespan—its ability to impede the permeation of methanol between the anode and cathode. The intrusion of methanol within the fuel cell milieu has the potential to disrupt redox reactions, resulting in a decrease in fuel cell efficiency. Moreover, this predicament could escalate to graver consequences, encompassing leakage, corrosion, or even perilous detonations.

The interplay between the methanol permeability of crosslinked membranes and their corresponding crosslinking densities was meticulously investigated via a binary diffusion cell, as graphically illustrated in Figure 12. Remarkably, a discernible congruity with the trends observed in the preceding section, concerning water molecule translocation, emerges. It is conceivable that the phenomenon of methanol permeation within the membrane may share noteworthy parallels with the transport of water molecules. Analogously, methanol molecules might navigate the polymer’s internal free volume and water pathways during the processes of infiltration and diffusion.

### 3.5. Proton Conductivity

As depicted in Figure 13a, the proton conductivity of membranes under full hydration conditions at 25 °C demonstrates a trend of initially increasing and subsequently decreasing with ascending crosslinking density. The inflection point that marks this transition consistently emerges at a crosslinking temperature of 125 °C. Our perspective retains the notion that the ramifications of crosslinking on proton conductivity are primarily twofold, including membrane crosslinking density and crystallinity. When the crosslinking temperature is below 125 °C, the dominance of crystallinity becomes evident in governing methanol transport. As expounded in the preceding section, reduced crystallinity facilitates water adsorption, intramembrane diffusion, and the augmentation of free volume. This, in turn, facilitates the ingress of water molecules into the membrane, fostering the creation of water clusters encircling phosphate groups, ultimately elevating the dissociation of these groups. Concurrently, the vehicular transport mechanism of protons and water molecules within the membrane collectively amplifies proton conductivity. In contrast, the scenario changes when crosslinking occurs at temperatures surpassing 125 °C. Here, the effect of crosslinking density outpaces the influence of crystallinity on proton conductivity. Heightened membrane crosslinking density engenders a more compact structure, hampering molecular chain mobility and impeding the diffusion of water molecules. Consequently, proton transport is suppressed. 

Nafion’s suboptimal proton conductivity in high-temperature, low-humidity conditions is a widely recognized phenomenon. As explicated in the preceding section, water molecule adsorption within the membrane profoundly affects its transport properties. Thus, the proton conductivity of PVA-based crosslinked membranes was explored under low-humidity conditions, as illustrated in Figure 13b. Intriguingly, the proton conductivity of the membrane displays a steady and consistent increase with escalating crosslinking density under such conditions. This trend diverges sharply from the behavior observed under full hydration. To unveil the underlying rationale, we scrutinized the proton conductivity of membranes with varying water content and two distinct crosslinking densities.

The linear-scale representation of proton conductivity against the volume fraction of water, as depicted in Figure 14, unveils the phenomenon of percolation, an observation shared with Nafion membranes. Specifically, a threshold value for proton conductivity is discernible, and once the water content surpasses this threshold, a swift enhancement in proton conductivity transpires. The PVA/PBTCA membrane with higher crosslinking density exhibits a lower percolation threshold (0.1) than its lower crosslinking density counterpart (0.2). This threshold is comparable to Nafion’s at 0.1. This divergence implies that membranes with lower crosslinking density necessitate a higher water content for efficient proton transportation. To expound on this, we introduce the concept of a competitive water absorption theory to elucidate the peculiar proton transport behavior witnessed under low-humidity conditions.

It is well-established that there exists only one hydrophilic group, the sulfonic acid group, in the Nafion membrane. However, alongside the hydrophilic phosphate groups, crosslinked PVA membranes feature a significant abundance of hydroxyl groups along molecular-chain side chains. In this context, upon water molecule infiltration into the membrane, water molecules not only adsorb around phosphate groups but also extensively engage in hydrogen bonding interactions with hydroxyl groups. This engenders a competitive scenario where water molecules become subject to a “tug-of-war” between phosphate groups and hydroxyl groups. This competition substantially diminishes the dissociation and conveyance of protons on phosphate groups, thereby diminishing the membrane’s proton transport efficiency. Heightened membrane crosslinking density diminishes the number of hydroxyl groups for crosslinking, consequently alleviating the adversarial effects of competitive water absorption on membrane performance.

The microstructural insight provided by Figure 15 showcases interconnected crosslinking nodes forming permeable water domains within the hydrated membrane, thus facilitating proton transport. Notably, the higher percolation threshold in PVA/PBTCA-80 suggests that a fraction of the permeable water domains contributes minimally to proton transport; this fraction results from water absorption by hydroxyl groups. Viewed from this perspective, an elevated membrane crosslinking density favors proton transport by reducing the proportion of water domains formed through hydroxyl groups’ absorption. However, in fully hydrated conditions, a greater degree of dissociation is exhibited by phosphate groups, dampening the competitive water absorption effect. Conversely, in relatively low-humidity conditions, owing to the pronounced hydrophilicity of hydroxyl groups, phosphate groups can merely “capture” a smaller fraction of water molecules, resulting in a marked reduction in dissociation. In this context, the competitive water absorption effect assumes greater prominence. Once the crosslinking temperature surpasses 125 °C, this notable competitive hydration phenomenon effectively mitigates the decline in conductivity, and engenders distinct proton conductivity behaviors under varying hydration states.

### 3.6. DMFC Performance

To assess the suitability of the fabricated crosslinked PVA membrane for DMFC applications, we conducted performance tests on the PVA/PBTCA-15-125 membrane at a temperature of 60 °C and relative humidity of 40%. As depicted in Figure 16, the open circuit voltage (OCV) of the fuel cell measures 0.78 V, signifying excellent air tightness and methanol-blocking capabilities of the membrane. Furthermore, it achieves the peak power density of 16.1 mW cm^−2^ at a current density of 81.3 mA cm^−2^.

## 4. Conclusions

In this work, we have successfully developed a novel phosphoric acid-containing crosslinked polyvinyl alcohol (PVA) membrane using a sol–gel approach, specifically tailored for application in proton exchange membrane fuel cells (PEMFCs). By ingeniously harnessing phosphoric acid’s dual role as a crosslinking agent and a source of hydrophilic -H_2_PO_3_ groups, we have engineered a membrane with enhanced properties for efficient proton conduction. Our investigation has unveiled a pivotal temperature threshold of 125 °C, where the crosslinking density becomes a determinant of performance. Remarkably, this critical temperature delineates a transition point at which proton conductivity and methanol permeability exhibit an intriguingly correlated behavior. Beyond this temperature, the delicate balance shifts, leading to a decline in these performance parameters with elevated PBTCA content. Central to these findings is the nuanced interaction between the membrane’s water absorption, diffusion characteristics, free volume, and crosslinking density. These interconnected structural attributes collectively dictate the membrane’s behavior within the challenging confines of fuel cell environments. As we unveil the intricate relationship between membrane architecture and functional properties, our study casts fresh light on the optimization of crosslinking conditions, offering a novel avenue to elevate the performance of proton exchange membranes in fuel cell applications. In conclusion, our work not only presents an innovative approach to synthesizing advanced PEMFC materials but also advances our understanding of the intricate interplay between membrane structure and performance. 

## Figures and Tables

**Figure 1 polymers-15-04198-f001:**
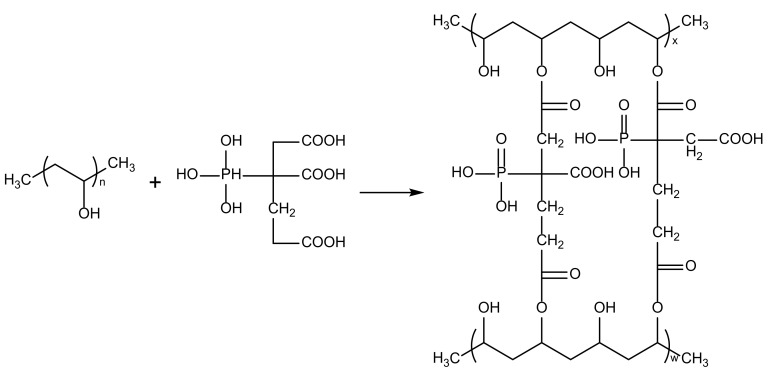
Possible reaction mechanism of PVA and PBTCA.

**Figure 2 polymers-15-04198-f002:**
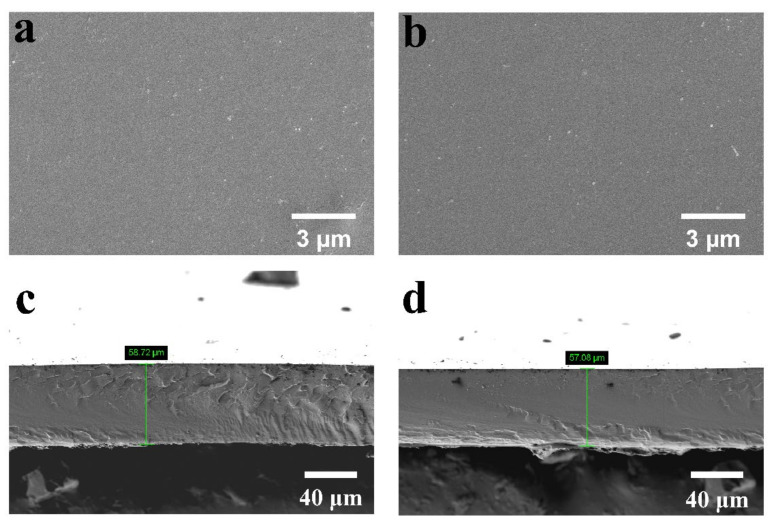
The surface and cross-section SEM images of PVA/PBTCA-5-125 (**a**,**c**) and PVA/PBTCA-30-125 (**b**,**d**).

**Figure 3 polymers-15-04198-f003:**
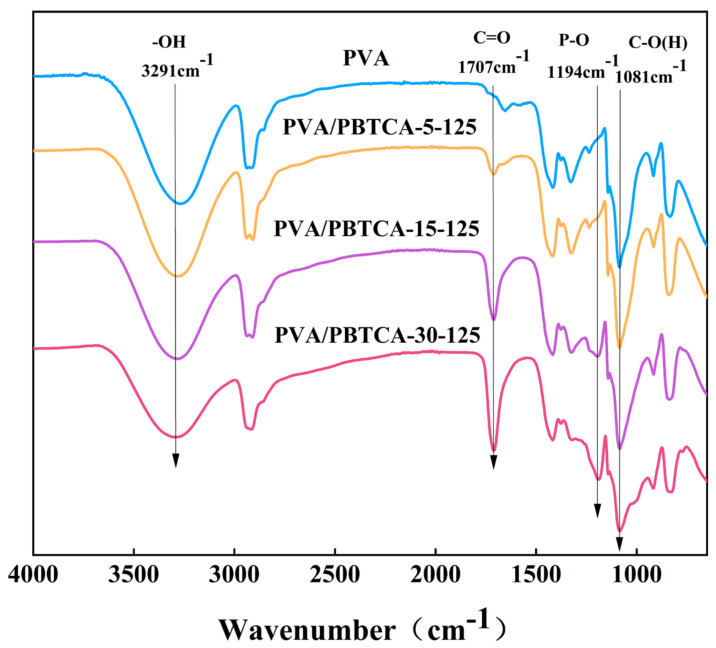
FT-IR spectra of the crosslinked PVA/PBTCA membranes: PVA, PVA/PBTCA-5-125, PVA/PBTCA-15-125, and PVA/PBTCA-30-125.

**Figure 4 polymers-15-04198-f004:**
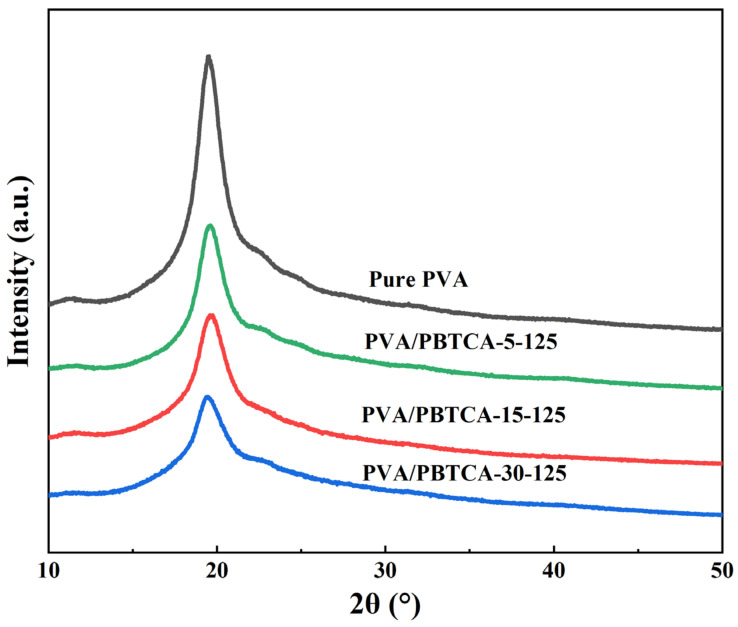
X-ray diffractograms of membranes.

**Figure 5 polymers-15-04198-f005:**
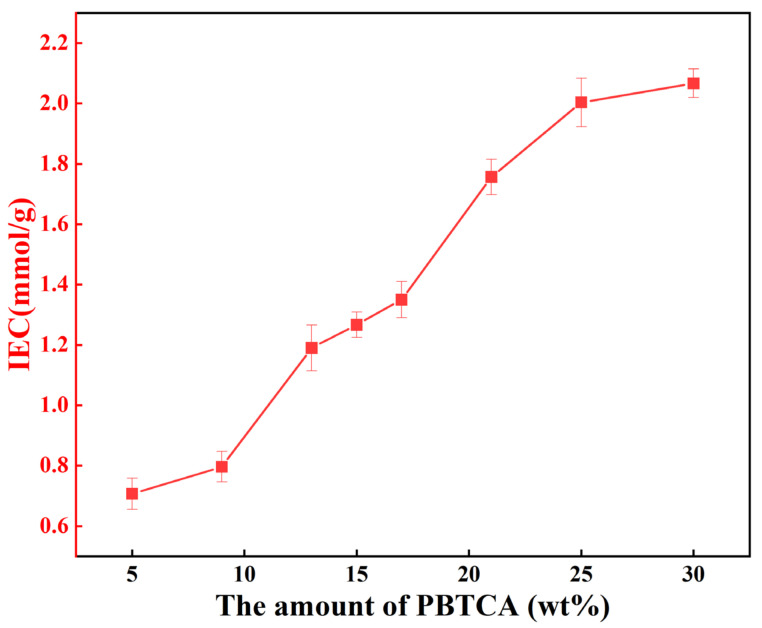
The change in IEC values in the PVA/PBTCA-125 membranes depending on the amount of PBTCA and the crosslinking temperature.

**Figure 6 polymers-15-04198-f006:**
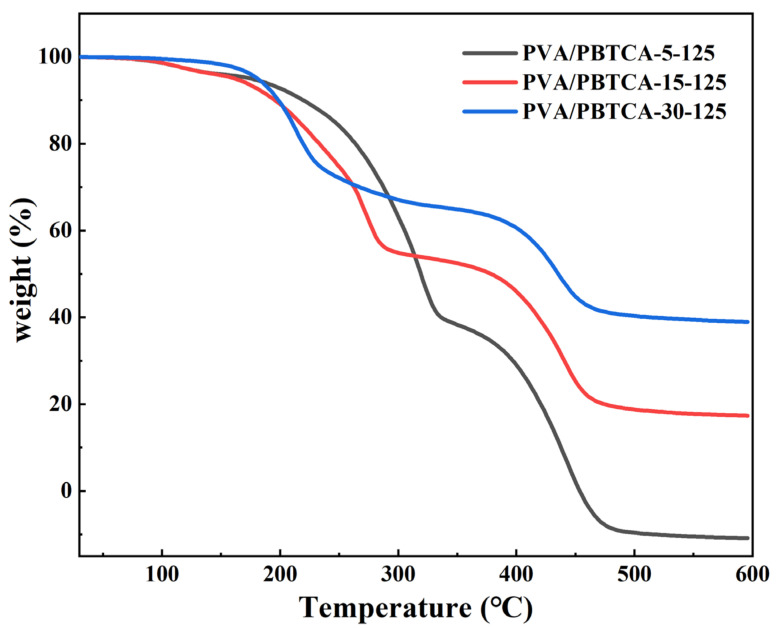
TGA thermograms of the crosslinked PVA/PBTCA membranes: PVA/PBTCA-125-5, PVA/PBTCA-125-15, and PVA/PBTCA-125-30.

**Figure 7 polymers-15-04198-f007:**
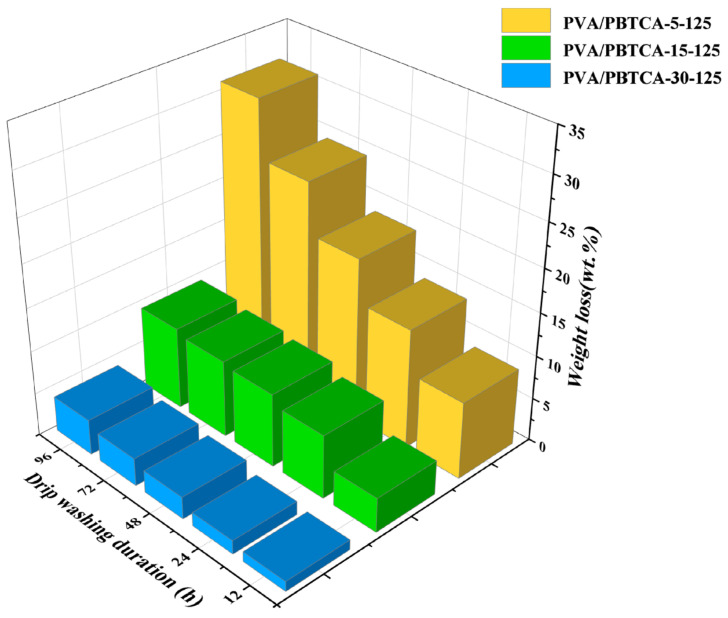
Weight losses of PVA-based films during drip-washing test at 80 °C for 4 days.

**Figure 8 polymers-15-04198-f008:**
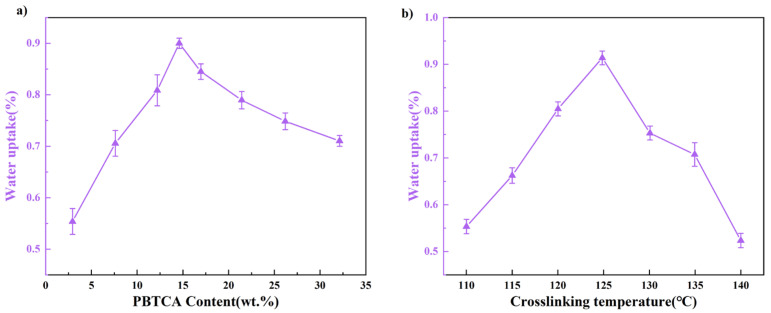
The change of water uptake in the PVA/PBTCA membranes depending on the amount of PBTCA (**a**) and the crosslinking temperature (**b**).

**Figure 9 polymers-15-04198-f009:**
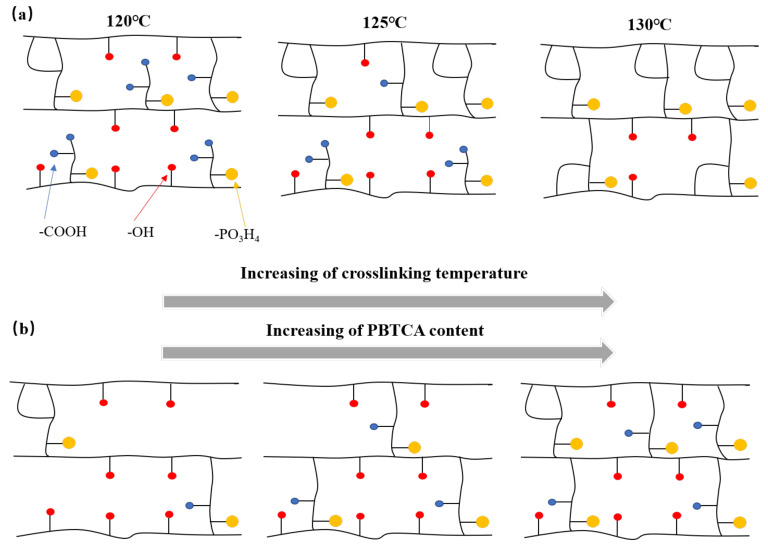
Postulated crosslinking mechanism of PVA/PBTCA; (**a**) the effect of crosslinking temperature and (**b**) the effect of the amount of PBTCA in the PVA.

**Figure 10 polymers-15-04198-f010:**
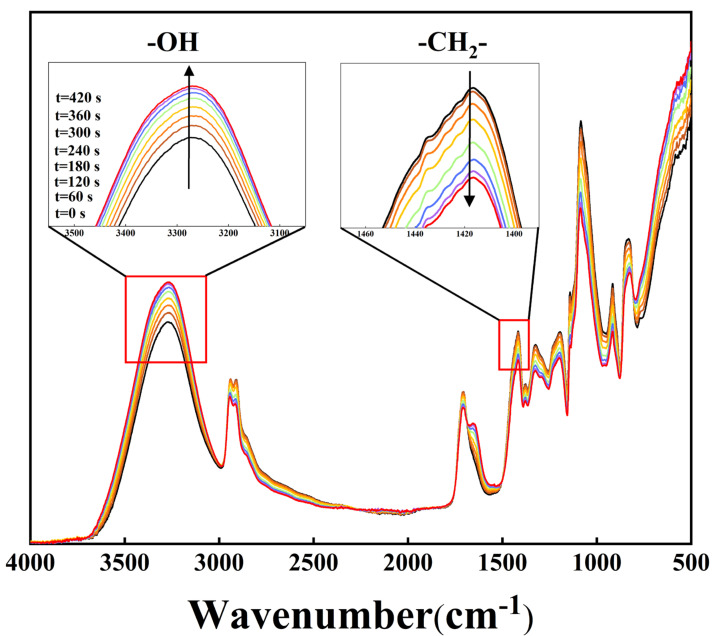
Infrared spectra of liquid water diffusing into dry PVA/PBTCA-15-125 at 25 °C over selected time intervals. The inset illustrates the increasing O-H bending band (water) and decreasing -CH_2_- stretching band (polymer) with time. Arrows indicate the direction of spectral changes over time.

**Figure 11 polymers-15-04198-f011:**
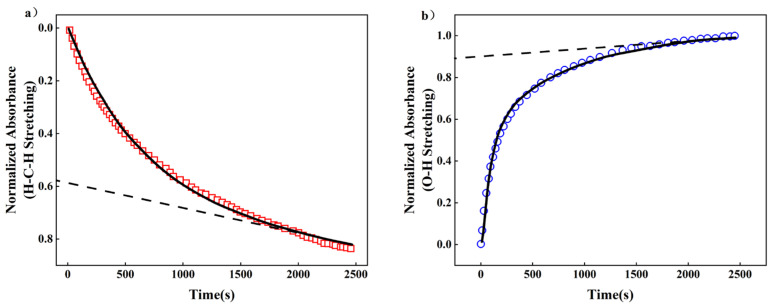
(**a**) The time-resolved, normalized -CH_2_- stretching absorbance data (red squares) of PVA/PBTCA-15-125 at 25 °C. The black line represents the optimal fit regression of the tri-modal viscoelastic relaxation model (Equation (15)), where the relaxation time constant, β, is the sole adjustable parameter. (**b**) Time-resolved, normalized O-H stretching absorbance data (blue circles) of PVA/PBTCA-15-125 at 25 °C. The solid black line represents the optimal fit regression of the diffusion relaxation model (Equation (16)).

**Figure 12 polymers-15-04198-f012:**
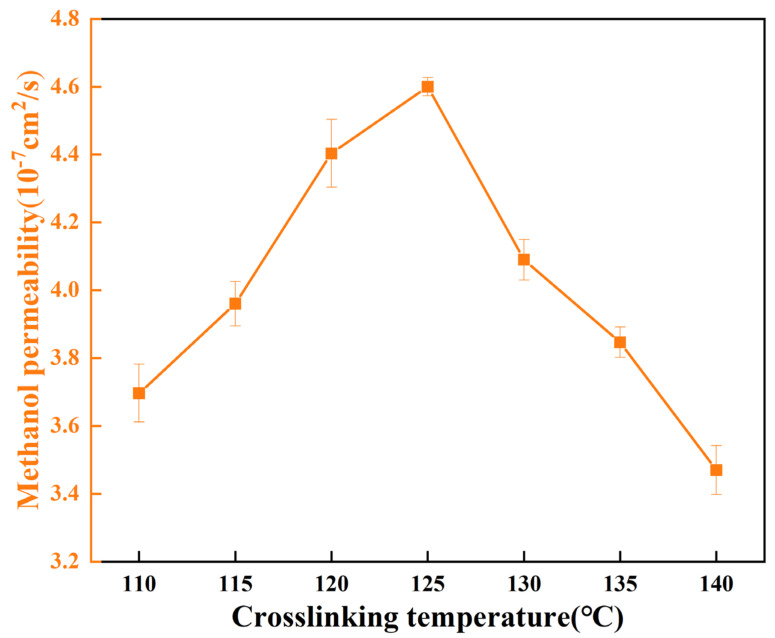
The methanol permeabilities of the PVA/PBTCA membranes measured at 25 °C (feed: 2 M methanol solution).

**Figure 13 polymers-15-04198-f013:**
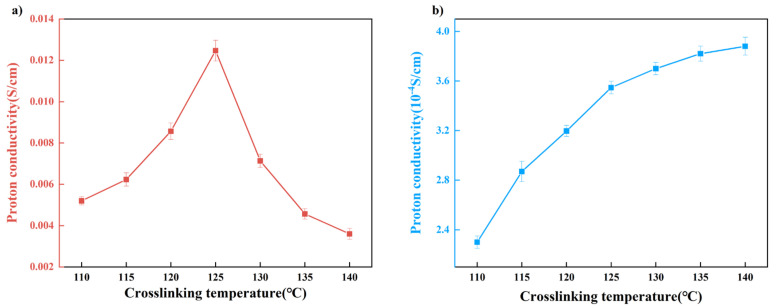
The proton conductivities of the PVA/PBTCA membranes measured under full−hydration conditions (**a**) and 70%RH (**b**) at 25 °C.

**Figure 14 polymers-15-04198-f014:**
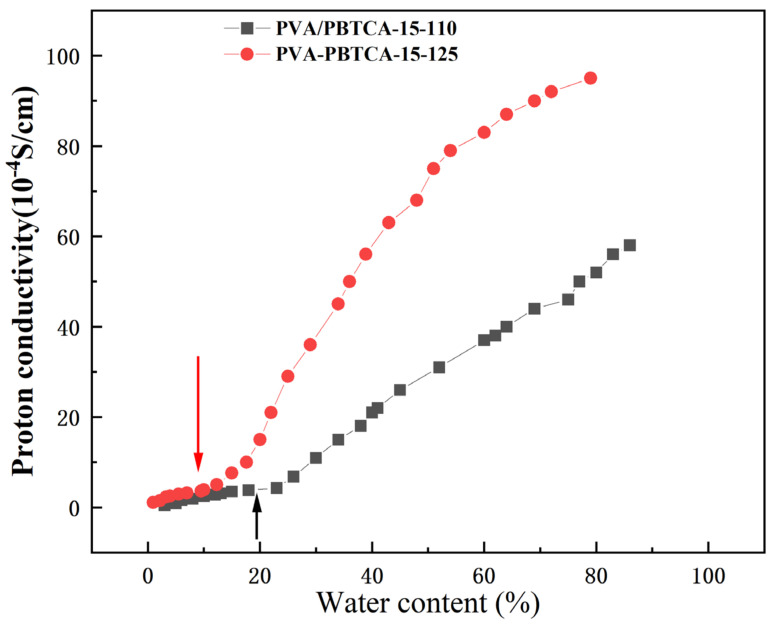
Variation in proton conductivity with volume fraction of water (arrow indicates percolation threshold).

**Figure 15 polymers-15-04198-f015:**
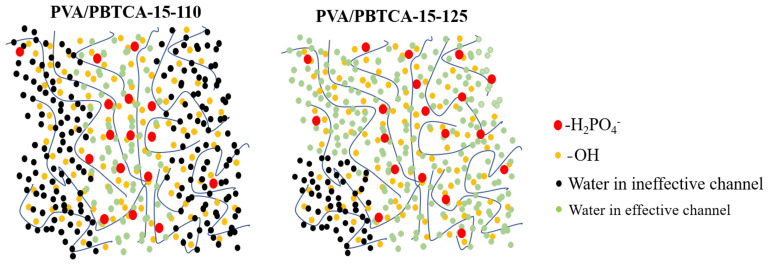
Hypothetical models of water domains of PVA/PBTCA-15-110 and PVA/PBTA-15-125.

**Figure 16 polymers-15-04198-f016:**
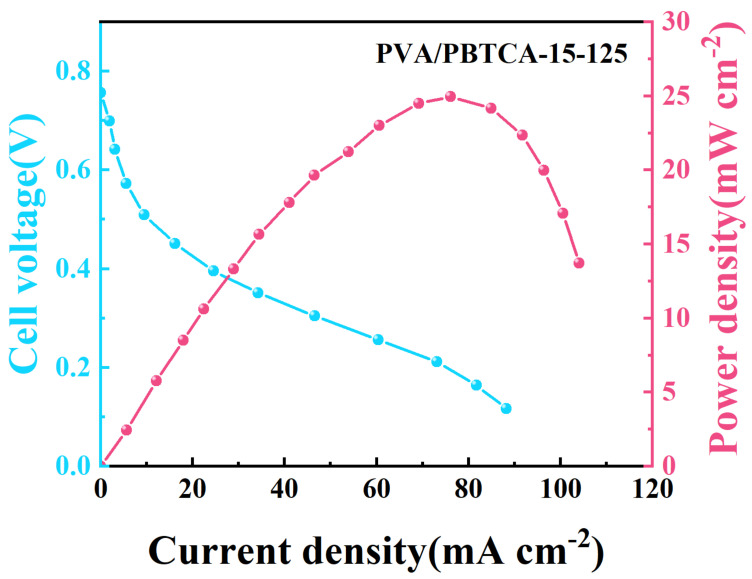
Polarization and power density curves of DMFCs using PVA/PBTCA-15-125 membrane (The light blue line is the polarization curve, and the rose red line is the power density curve).

**Table 1 polymers-15-04198-t001:** The degree of crosslinking and crystallinity of PVA-PBTCA membranes.

Sample Code	Degree of Crystallinity (%)	Crosslinking Density (mol/m^3^)
PVA/PBTCA-5-110	41.89	810.8 ± 76.8
PVA/PBTCA-15-110	38.12	950.3 ± 58.9
PVA/PBTCA-30-110	29.53	1201.8 ± 45.9
PVA/PBTCA-5-125	39.77	1019.6 ± 101.7
PVA/PBTCA-15-125	35.52	1325.9 ± 120.6
PVA/PBTCA-30-125	26.14	1539.8 ± 186.0
PVA/PBTCA-5-140	35.48	1591.2 ± 84.1
PVA/PBTCA-15-140	22.34	1856.2 ± 83.6
PVA/PBTCA-30-140	11.56	2250.6 ± 104.4

**Table 2 polymers-15-04198-t002:** Summary of the positron lifetime data of PVA/PBTCA membranes.

Samples	τ_1_ (ns)	τ_2_ (ns)	τ_3_ (ns)	I_1_ (%)	I_2_ (%)	I_3_ (%)	R_PALS_ (nm)	V_PALS_ (nm^3^)
PVA/PBTCA-15-110	0.186	0.422	1.351	41.064	45.876	13.059	0.215	0.0416
PVA/PBTCA-15-120	0.189	0.439	1.363	44.223	43.000	12.778	0.217	0.0428
PVA/PBTCA-15-125	0.188	0.424	1.392	42.528	46.877	10.595	0.221	0.0452
PVA/PBTCA-15-130	0.182	0.437	1.365	42.580	42.828	14.592	0.218	0.0434
PVA/PBTCA-15-140	0.181	0.416	1.338	40.074	46.784	13.142	0.214	0.0410

**Table 3 polymers-15-04198-t003:** Relaxation time constants and water diffusion coefficients for crosslinked PVA membranes.

Sample Code	Thickness (μm)	Relaxation Time Constants (β, 10^−5^ s^−1^)	Water Diffusion Coefficients (D, 10^−7^ cm^2^ s^−1^)
PVA/PBTCA-15-110	68.5	1.3	4.4
PVA/PBTCA-15-120	70.2	4.5	9.5
PVA/PBTCA-15-125	68.3	9.2	12.8
PVA/PBTCA-15-130	67.9	6.4	8.6
PVA/PBTCA-15-140	69.4	5.2	3.1

## Data Availability

Not applicable.

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
