# Peer review of "Effect of Crosslinking Conditions on the Transport of Protons and Methanol in Crosslinked Polyvinyl Alcohol Membranes Containing the Phosphoric Acid Group"

_polymers, 2023, doi:10.3390/polym15214198_

Round 1

Reviewer 1 Report (Previous Reviewer 2)

Authors have addressed my comments in the resubmission.

Author Response

Reviewer 2 Report (New Reviewer)

The presented manuscript is devoted to the use of polyvinyl alcohol for the formation of membranes. The authors of the work, in their theoretical part, adequately explain what membranes currently exist (Nafion), their advantages and disadvantages, and why it makes sense to use PVA to obtain new materials with the required transport properties. Methods for modifying PVA through the use of sulfosuccinic acid to regulate the proton conductivity of a PVA membrane are described. The role of phosphoric acid in the creation of a dynamic network of hydrogen bonds is explained. As a result, the authors lead the reader to the main goal of the work - the use of phosphoric acid to modify PVA in order to obtain a polymer electrolyte membrane.

In the work, the authors use a wide range of approaches to prove that the degree of cross-linking of PVA changes with changes in PBTCA content, ranging from IR spectroscopy to X-ray diffraction analysis. These patterns can be considered in the future as basic. Those, when assessing the structure of cross-linked systems, the degree of cross-linking can be roughly assessed. The influence of PBTCA on water uptake shows that up to 15% the authors observe an increase in values, with a further increase in PBTCA values they decrease. The authors explain this phenomenon well from the point of view of changes in structure in the system and then the resulting changes are presented schematically, and here, in my opinion, the presented explanations are lost because the change in order (its deterioration) in the system is not visible!?

Line 14. It is better to use "PA" instead of "PBTCA" for phosphoric acid.

Line 76. "as documented by R. Chodankar et al" - I recommend deleting.

2.2. Membrane preparation. I'm not sure that assessing the quality of a solution by its transparency is sufficient. It is advisable to evaluate the homogeneity of the solution using optical microscopy. It is known that even at high temperatures, PVA solutions with a concentration of more than 6% can retain supramolecular structures in a system with water.

Why were systems with PBTCA produced at 50°C? On what basis did you choose the time for keeping the films in the oven?

The description of transverse chips in Figure 2 needs to be given more attention.

The conclusions of the work contain the main results achieved and are distinguished by their brevity.

In general, the work deserves attention and can be published in the journal Polymers.

Author Response

This manuscript is a resubmission of an earlier submission. The following is a list of the peer review reports and author responses from that submission.

Round 1

Reviewer 1 Report

This is an interesting work reports the adequate characterization of cross-linked PVA polymers with phosphoric acid groups. The authors used multi-characterization techniques to establish the proton dynamics and permeability of methanol in the studied polymer membranes. Specifically, the effect of phosphoric acid concentration and temperature on the cross-linking phenomenon and proton mobility have been thoroughly investigated. In my opinion, this work is very interesting to the readers of the journal Polymers and I would recommend it to be accepted with minor revisions. Following minor comments should be considered to further improve the manuscript.

1. Experimental details of SEM characterization is missing.

2. Double-check the sample labels in Figure 3 caption and Table 1. The designation of samples as PVA/PBTCA-X-Y should consistent all over the manuscript.

3. I would recommend authors to discuss the analysis of free and bound water using FTIR spectrum.

4. Does the diffusion coefficient shown in Table 4 signify the mobility of only bound water or the average of both free and bound water? Please elaborate this in the discussion.

5. Page 14, line 438: “the water absorption rate initiated a gradual ascent”, isn’t it supposed to be “descent”?

6. Page 19, line 558-560: please double-check the sentence.

7. Please comment on Figure 13b (conductivity v/s temperature plot) in the discussion.

Reviewer 2 Report

In this paper, the authors report on the synthesis and application of polyvinyl alcohol (PVA) membranes and their versatile properties. The membranes are designed as cross-linked PVA membranes using phosphoric acid (PBTCA) as the cross-linking agent and proton transfer medium, and the samples are used to study the effect of temperature. The optimum membrane film is reported to have proton conductivities between 10-3 and 10-2 S/cm and methanol permeabilities between 10-8 and 10-7 cm²/s.

1. The manuscript needs to be rewritten by the authors. In the experimental section, many sentences and paragraphs have been copied and pasted from a published paper (Preparation and characterization of PVA proton exchange membranes containing phosphonic acid groups for direct methanol fuel cell applications). The DSC measurement and the experimental steps are copied and pasted from this paper (Preparation and characterization of crosslinked PVA/SiO2 hybrid membranes containing sulfonic acid groups for direct methanol fuel cell applications). The information is 100% identical. The equations used to calculate the data are very similar to this paper (Anomalous, Multistage Liquid Water Diffusion and Ionomer Swelling Kinetics in Nafion and Nafion Nanocomposites). Plagiarism should not be accepted in academic research. The discussion of the faction of free water copies and pastes the discussion of Qmelting from this paper (Crosslinked poly(vinyl alcohol) membranes containing sulfonic acid group: proton and methanol transport through membranes).

2. In the experimental, the Methanol Permeability Test section should be 2.3.7. Then section 2.4 is missing.

3. The thickness of the membranes should be analysed by cross-sectional SEM.

4. The FTIR discussion and the FTIR peaks do not agree with the data shown in Figure 3.

5. The XRD peaks are reported at the same 19.8 degree in a similar sentence reported in this paper (Preparation and characterization of PVA proton exchange membranes containing phosphonic acid groups for direct methanol fuel cell applications)

6. PEMFC should be tested to prove the comments made about the membranes.

7. Morphological analysis is proposed to determine the membrane structure. EIS analysis is proposed to support the charge transport mechanism.

8. The Postulated crosslinking mechanism of PVA/PBTCA is highly similar to this paper (Crosslinked poly(vinyl alcohol) membranes containing sulfonic acid group: proton and methanol transport through membranes)
